# Dispositional mindfulness and mental health in Chinese emerging adults: A multilevel model with emotion dysregulation as a mediator

**Rebecca Y. M. Cheung**[1], **Zijun Ke**[2]*, **Melody C. Y. Ng**[3]

**1** Department of Early Childhood Education, Centre for Child and Family Studies, and Centre for Psychosocial Health, The Education University of Hong Kong, Hong Kong SAR, China, **2** Department of Psychology, Sun Yat-sen University, Guangzhou, Guangdong, China, **3** Department of Psychology, The Chinese University of Hong Kong, Hong Kong SAR, China

* keziyun@mail.sysu.edu.cn

**Data Availability Statement:** All relevant data are within the paper and its Supporting information files.

## Abstract

Using a multilevel model, this study examined emotion dysregulation as a mediator between dispositional mindfulness and mental health among Chinese emerging adults. Participants were 191 Chinese emerging adults (female = 172) between 18 and 27 years old ($M$ = 21.06 years, $SD$ = 2.01 years), who completed a questionnaire that assessed their dispositional mindfulness, emotion dysregulation, and mental health outcomes for three times over 12 months, with a three-month lag between each time point. Within-person analysis revealed that emotion dysregulation mediated between dispositional mindfulness and mental health outcomes, including subjective well-being and symptoms of depression and anxiety. Time was positively associated with emotion dysregulation and negatively associated with symptoms of depression and anxiety. Between-person analysis revealed that emotion dysregulation negatively mediated between dispositional mindfulness and symptoms of depression and anxiety, but not subjective well-being. These findings call attention to within-person versus between-person effects of emotion dysregulation as a mediator between dispositional mindfulness and psychological outcomes, particularly of symptoms of depression and anxiety. Attesting to the relations established in western societies, the relations are also applicable to emerging adults in the Chinese context. Evidence was thus advanced to inform translational research efforts that promote mindfulness and emotion regulation as assets of mental health.

## Introduction

Mindfulness is a mental state whereby individuals attend to their cognitive, physical, and emotional experience in the present moment nonjudgmentally [1, 2]. Previous research suggested that mindfulness is associated with mental health outcomes, such as a low level of psychological symptoms and better subjective well-being [3–6]. Moving beyond the simple association

**Funding:** "This work was supported by the National Natural Science Foundation of China [31700986] and The Education University of Hong Kong [RG 60/2014-2015R]."

**Competing interests:** The authors have declared that no competing interests exist.

between mindfulness and mental health, recent studies have identified several mediating mechanisms, including greater self-esteem [7], greater positive affect, hope, and optimism [8], lower rumination [9], and greater emotion regulation [10, 11], thereby suggesting possible chains of processes between mindfulness and mental health. Among these mediators, emotion regulation has received much theoretical and empirical attention in recent years [12, 13].

Emotion regulation is defined as a process in modulating emotions and emotional responses [14, 15]. According to Gratz and Roemer [16], adaptive emotion regulation involves such aspects as acceptance of emotional responses, engagement in goal directed behavior, maintenance of emotional clarity and awareness, and access to adaptive emotion regulation strategies, such as cognitive reappraisal and an ability to savor positive experiences. On the contrary, emotion dysregulation involves difficulties in acquiring regulatory skills or a frequent use of maladaptive strategies, such as rumination, avoidance, and expressive suppression [17, 18]. Previous research suggested that our capability to regulate emotions is central to subjective well-being [18]. Moreover, emotion dysregulation not only undermines well-being, but also gives rise to clinical depression, anxiety, psychological distress, and elevated depressive and anxiety symptoms [10, 17, 19–21].

## Theoretical tenets of emotion regulation as a mediator between mindfulness and mental health outcomes

As a proximal correlate of psychological well-being, people's capability to regulate emotions is associated with mindfulness [6, 22, 23]. More specifically, Teper et al. [13] theorized that mindfulness is linked to emotion regulation through heightened metacognitive awareness, non-judgmental acceptance, and executive control, all of which are crucial to well-being. Likewise, in mindfulness-to-meaning theory, Garland et al. [12] postulated that mindfulness promotes purpose in life by virtue of greater metacognitive awareness, broadened attention to context, adaptive emotion regulation, ability to savor hedonic experience, and prosocial actions. Both theoretical frameworks suggest close connections between mindfulness, emotion regulation, and psychological well-being.

Supporting the frameworks, cross-sectional studies revealed that emotion regulation mediated between dispositional mindfulness and lower levels of depression and anxiety [11, 23–25] as well as better life satisfaction [26]. Studies based on a clinically depressed and anxious sample similarly suggested that emotion regulation strategies, including cognitive reappraisal and rumination, mediated between dispositional mindfulness and symptoms of depression and anxiety [25]. These findings resonated with research conducted with non-clinical emerging adult samples, in that emotion regulation mediated the inverse relation between dispositional mindfulness and depressive symptoms, in that dispositional mindfulness was related to better emotion regulation and fewer depressive symptoms [23]. In a similar vein, Coffey and Hartman [11] and Coffey et al. [24] found that emerging adults' regulation of negative emotions, impulse control, and goal engagement mediated between dispositional mindfulness and well-being. Altogether, evidence to-date converged to indicate emotion regulation as a vital process between dispositional mindfulness and well-being.

## Processes of mental health in the Chinese context

Among the handful of studies conducted in the Chinese context, dispositional mindfulness predicted better impulse control, better emotion regulation, less procrastination, and less psychological distress [10, 27]. Moreover, mindful awareness was associated with subsequent changes of stress response, emotion regulation, and anxiety symptoms [28, 29]. Changes in emotion regulation strategies were also associated with changes in depressive symptoms, life

satisfaction, and general health over time [30]. Consequently, timing and changes were crucial in linking between mindfulness, emotion regulation, and mental health outcomes. Beyond dispositional mindfulness, mindfulness training also increased Chinese adolescents' and adults' well-being and psychological adjustment in Hong Kong [31, 32]. In a randomized controlled trial involving a Chinese sample with clinical anxiety, Wong et al. [33] revealed that participants who took an 8-week mindfulness-based cognitive therapy experienced increased dispositional mindfulness and reduced anxiety. In addition, such an increase was found to be more significant in these participants than did participants who received psychoeducation based on cognitive behavioral therapy principles. Similar findings were reported in another randomized controlled trial involving a Chinese sample with recurrent symptoms of depression and anxiety [34]. In the study, participants who received an 8-week compassion–mindfulness therapy showed more mental health improvements than did participants from the waitlist control group. These findings reveal that the mechanisms associated with mindfulness are integral to mental health in the Chinese context.

## Within- versus between-person effects

A majority of studies to-date focus on between-person associations of mindfulness, emotion regulation or dysregulation, and mental health [e.g., 5–11]. Only a handful of studies have partitioned within-person from between-person processes. Between- and within-person effects are conceptually and statistically independent. Within-person effect aids the understanding of *intraindividual* processes underlying well-being. For instance, when mindfulness of person *i* increases, then his or her own mental health is expected to increase over time. Between-person effect accounts for *interindividual* processes underlying well-being. That is, when person *i* has a higher score in mindfulness than person *j*, then person *i* is expected to have a higher score in mental health than is person *j*. Of note, partitioning among within- and between-person associations is important, because otherwise the results could sometimes be biased [35] or uninterpretable [36].

Several studies to-date have demonstrated the utility of within-person analyses in mindfulness and well-being. For example, Galla [37] indicated that a five-day mindfulness training was associated with significant within-person reductions in rumination and depressive symptoms, as well as increases in life satisfaction in healthy adolescents. In addition, many of the improvements were maintained at the three-month follow-up assessment. In another study, an 8-week mindfulness training gave rise to an inverse within-person association between mindfulness and relationship stress [38]. Apart from stress, psychological symptoms, and subjective well-being, another study also showed within-person associations between dispositional mindfulness and relationship satisfaction [39]. By investigating within-person effects, these studies methodologically precluded between-person effects of mindfulness on psychological outcomes.

## This study

Aside from the studies described in the previous section, past research predominantly examined between-person effects among mindfulness, emotion regulation, and mental health [5–11]. Although some studies have begun to examine within-person effects [37–39], few, if any, have teased apart within-person from between-person effects. Although within- and between-person findings converged to suggest the mental health benefits of mindfulness, it remains unclear as to whether its significance on emotion regulation and various mental health outcomes differ.

The present study investigated longitudinal within- versus between-person mediating effects of dispositional mindfulness and mental health outcomes via emotion dysregulation. Through this analytic technique, the intraindividual and interindividual processes could be identified. Based on the literature [23–25], we hypothesized that greater dispositional mindfulness would be related to lower emotion dysregulation. Lower emotion dysregulation would, then, be related to fewer symptoms of depression and anxiety, as well as greater subjective well-being. We further hypothesized that the relation between dispositional mindfulness and mental health outcomes would be mediated by emotion dysregulation at both within- and between-person levels. The strength of associations may vary as a function of specific levels. As stated earlier, time was an important measure in assessing changes within and between variables [28, 30]. Consequently, time was included as a within-person covariate to control for dispositional mindfulness, emotion dysregulation, and mental health outcomes. Based on previous research showing the links between gender and depressive symptoms [40], anxiety symptoms [41], and subjective well-being [42], gender was included as a between-person covariate for mental health outcomes.

## Method

### Participants

Participants were 191 Chinese emerging adults (female = 172; 90.05%) recruited online through two mass emails at a major public university in Hong Kong. Inclusion criteria included university-enrolled emerging adults who were proficient in Chinese, within the age range between 18 and 29 years old [43] and agreed to participate for three time points over the course of a year. Participants were between 18 and 27 years old, with a mean age of 21.06 years at Time 1 ($SD$ = 2.01 years). As for the retention rate, 93.72% ($n$ = 179) of the participants from Time 1 (T1) were retained at Time 2 (T2); 94.41% ($n$ = 169) of the participants from T2 were retained at Time 3 (T3).

### Procedures

The study was approved by the Human Research Ethics Committee of The Education University of Hong Kong (Approval #: 2015-2016-0352) prior to its implementation. All procedures performed were in accordance with the ethical standards of the institutional research committee and the 1964 Helsinki declaration and its later amendments. A written consent was obtained from each participant prior to the beginning of the study. At the first time point, participants completed the baseline measures. They were then invited to complete the follow-up questionnaires twice, with a three-month lag between time points, over the course of a year. Each packet took approximately 30 minutes to complete. Upon completion, participants received a supermarket coupon at each time point (totaling HK$200, i.e., ~US$25.71).

### Measures

**Dispositional mindfulness.** The 12-item Cognitive and Affective Mindfulness Scale–Revised (CAMS-R) [44] was used to assess dispositional mindfulness on a 4-point scale from 1 (*rarely/ not at all*) to 4 (*almost always*). The CAMS-R assessed three dimensions of mindfulness including awareness, acceptance, and attention. A sample item included, "I can usually describe how I feel at the moment in considerable detail." The item scores were averaged, such that higher scores indicated a greater level of mindfulness. The measure was validated previously in a Chinese college sample with adequate convergent validity, reliability, and factor structure [45]. Specifically, participants from this study had a similar mean level of

**Table 1. Descriptive statistics of the variables under study (*N* = 191).**

| Variable | *M* | *SD* | Min | Max | Range of Scale | Intraclass Correlation | Design Effect |
|---|---|---|---|---|---|---|---|
| (1) Gender (0 = male; 1 = female) | - | - | | | | | |
| **Time 1** | | | | | | | |
| (2) Dispositional mindfulness | 2.85 | .43 | 1.80 | 4.00 | 1.00–4.00 | | |
| (3) Emotion dysregulation | 2.03 | .49 | 1.15 | 3.66 | 1.00–5.00 | | |
| (4) Depressive symptoms | .61 | .50 | .00 | 3.00 | .00–3.00 | | |
| (5) Anxiety symptoms | .61 | .58 | .00 | 3.00 | .00–3.00 | | |
| (6) Subjective well-being | 4.49 | .86 | .79 | 5.00 | .00–5.00 | | |
| **Time 2** | | | | | | | |
| (7) Dispositional mindfulness | 2.80 | .48 | 1.60 | 4.00 | 1.00–4.00 | | |
| (8) Emotion dysregulation | 2.12 | .51 | 1.17 | 3.47 | 1.00–5.00 | | |
| (9) Depressive symptoms | .68 | .56 | .00 | 3.00 | .00–3.00 | | |
| (10) Anxiety symptoms | .65 | .64 | .00 | 3.00 | .00–3.00 | | |
| (11) Subjective well-being | 4.28 | .86 | .50 | 5.00 | .00–5.00 | | |
| **Time 3** | | | | | | | |
| (12) Dispositional mindfulness | 2.78 | .52 | 1.20 | 4.00 | 1.00–4.00 | | |
| (13) Emotion dysregulation | 2.12 | .54 | 1.16 | 3.93 | 1.00–5.00 | | |
| (14) Depressive symptoms | .74 | .61 | .00 | 2.89 | .00–3.00 | | |
| (15) Anxiety symptoms | .81 | .68 | .00 | 3.00 | .00–3.00 | | |
| (16) Subjective well-being | 4.15 | .92 | .14 | 5.00 | .00–5.00 | | |
| **Across Time Points** | | | | | | | |
| Dispositional mindfulness | 2.81 | .48 | | | 1.00–4.00 | .56*** | 2.01 |
| Emotion dysregulation | 2.09 | .51 | | | 1.00–5.00 | .73*** | 2.31 |
| Depressive symptoms | .68 | .56 | | | .00–3.00 | .64*** | 2.13 |
| Anxiety symptoms | .68 | .63 | | | .00–3.00 | .51*** | 1.93 |
| Subjective well-being | 4.32 | .89 | | | .00–5.00 | .63*** | 2.14 |

*Note*:

***$p < .001$. Design effect was defined as 1 + (average cluster size—1) × intraclass correlation.

dispositional mindfulness (see Table 1). In this study, the test-retest correlations of the measure between T1, T2, and T3 were moderate at .53—.58, $ps < .001$. Cronbach's alpha of the measure = .91—.94 between T1 and T3. More specifically, between T1 and T3, Cronbach's alpha of the awareness subscale = .63—.80, the acceptance subscale = .62—.79, and the attention subscale = .77—.81.

**Emotion dysregulation.** The 36-item Difficulties in Emotional Regulation Scale (DERS) [16] was used to assess emotional regulation on a 5-point scale from 1 (*almost never*) to 5 (*almost always*). The measure included 6 subscales including non-acceptance of emotions, difficulties in engaging in goal-directed behavior, impulse control difficulties, lack of emotional awareness, limited access to emotion regulation strategies, and lack of emotional clarity. Sample items included, "I am clear about my feelings" and "When I'm upset, I believe that my feelings are valid and important." The final scores were then averaged, such that higher scores indicated a greater emotion dysregulation. The measure had been validated in a Chinese sample [46]. Compared to another Chinese adult sample [47], participants from this study had a similar mean level of emotion dysregulation (see Table 1). In this study, the test-retest correlations of the measure between T1, T2, and T3 were .70—.76, $ps < .001$. Cronbach's alpha of the measure = .83—.87 between T1 and T3. More specifically, between T1 and T3, Cronbach's alpha of the non-acceptance of emotions subscale = .83—.87, the difficulties in engaging in

goal-directed behavior subscale = .56—.75, the impulse control difficulties subscale = .77—.85, the lack of emotional awareness subscale = .72—.75, the limited access to emotion regulation strategies subscales = .83—.87, and the lack of emotional clarity subscale = .74—.77.

**Depressive symptoms.**   The 9-item Patient Health Questionnaire-9 (PHQ-9) [48] was used to measure depressive symptoms in the past two weeks on a 4-point scale from 0 (*not at all*) to 3 (*nearly every day*). A sample item included, "Feeling bad about yourself—or that you are a failure or have let yourself or your family down?" The item scores were averaged to form a score of depressive symptoms, with higher scores indicating more symptoms. The measure has previously been validated in a Chinese community sample and demonstrated an adequate factor structure, construct validity, internal consistency, and test-retest reliability [49]. Compared to a cutoff score of 11 for the detection of depressive disorders [50], upon rescaling our mean scores to summed scores, participants' scores of 5.44–6.67 between T1 and T3 were below the clinical cutoff. Compared to another Chinese college sample [51] participants from this study had a comparable level of depressive symptoms (see Table 1). However, a total of 20, 21, and 28 participants at T1, T2, and T3, respectively, reported a summed score of above 11 (i.e., above the cutoff). In this study, the test-retest correlations of the measure between T1, T2, and T3 were .62—.67, *p*s < .001. Cronbach's alpha of the measure = .87—.91 between T1 and T3.

**Anxiety symptoms.**   The 7-item Generalized Anxiety Disorder-7 measure (GAD-7) [52] was used to measure anxiety symptoms in the past two weeks on a 4-point scale from 0 (*not at all*) to 3 (*nearly every day*). A sample item included, "Feeling nervous, anxious or on edge." The item scores were averaged to form a score of anxiety symptoms, with higher scores indicating more symptoms. The measure has been validated in a Chinese sample of hospital outpatients [53] and a sample of Chinese individuals with epilepsy [54]. Compared to a cutoff score of 10 for the detection of generalized anxiety disorder [52], upon rescaling our mean scores to summed scores, participants' scores of 4.22–5.67 between T1 and T3 were below the clinical cutoff. Compared to another Chinese college sample [51] participants from this study reported a similar level of anxiety symptoms (see Table 1). However, a total of 14, 20, and 26 participants at T1, T2, and T3, respectively, reported a total score of above 10 (i.e., above the cutoff). In this study, the test-retest correlations of the measure between T1, T2, and T3 were .52—.56, *p*s < .001. Cronbach's alpha of the measure = 93—.95 between T1 and T3.

**Subjective well-being.**   The 14-item Mental Health Continuum Short Form (MHC-SF) [55] was used as a measure for well-being over the past four weeks. A 6-point scale ranging from 0 (*never*) to 5 (*every day*) was used, with sample items including, "how often did you feel satisfied with life" (*emotional well-being*), "how often did you feel that that you had something important to contribute to society (*social well-being*) and "how often did you feel that your life has a sense of direction or meaning to it" (*psychological well-being*). The item scores were averaged to form a mean score, with higher scores indicating better well-being. MHC-SF had been previously validated in a Chinese adolescent sample and yielded good validity and reliability [56]. Compared to another Chinese college sample [57], participants from this study had a greater mean level of subjective well-being (see Table 1). In this study, the test-retest correlations of the measure between T1, T2, and T3 were .59—.69, *p*s < .001. Cronbach's alpha of the measure = .79—.94 between T1 and T3. More specifically, between T1 and T3, Cronbach's alpha of the emotional well-being subscale = .90—.94, the social well-being subscale = .79 —.86, and the psychological well-being subscale = .92—.93.

## Data analysis

Multilevel mediation analyses were conducted using Mplus 8.0 [58] to examine the mediating effect of emotion dysregulation on the relationship between dispositional mindfulness and

mental health outcomes, including depressive symptoms, anxiety symptoms, and subjective well-being, at both within- and between-person levels. To illustrate the necessity of multilevel modeling, we computed the intraclass correlations (ICC), which quantified the proportion of variance of the variables attributable to individual differences, as well as the design effect, defined as 1 + (average cluster size—1) × ICC [59]. Multilevel modeling has been used in previous research for panel data (e.g., [60]), with a minimum of three waves of observations to partition between within- and between-person associations, which, as introduced earlier, have distinctive practical implications. Compared to the traditional multilevel modeling approach to multilevel mediation, the multilevel structural equation modeling (MSEM) approach has the advantage of having a smaller bias.

In this study, we relied on the MSEM approach. Covariates at both levels were considered. At the within-person level, a time variable was included to control for the potential changes over time [61]. At the between-person level, gender was included to control for possible gender differences in symptoms of anxiety and depression. Regarding whether random slopes should be considered, with only three measurement occasions at the within-person level, the model allowing all mediation paths (i.e., "a" path from mindfulness to emotion dysregulation, "b" paths from mindfulness to mental health outcomes, and "c" paths from emotion dysregulation to mental health outcomes) and the slope of time to be random could not be identified. Therefore, we tested possible random mediation paths sequentially. Results showed that none of the "a", "b", and "c′" paths showed substantial individual differences, except for the "c′" path for depression. Additionally, we tested possible random slopes of time (i.e., change between two consecutive measurement occasions) and found no significant individual differences. Consequently, all mediation paths and slopes of time were fixed to be invariant across participants, except for the "c′" path for depression in the subsequent analyses.

In sum, the multilevel mediation model under consideration was as follows. The independent variable, the mediator, and the three dependent variables were partitioned into two latent parts:

$$\text{Dispositional Mindfulness}_{it} = \eta_{\text{Dispositional Mindfulness},it} + \eta_{\text{Dispositional Mindfulness},i}$$

$$\text{Emotion Dysregulation}_{it} = \eta_{\text{Emotion Dysregulation},it} + \eta_{\text{Emotion Dysregulation},i}$$

$$\text{Depression}_{it} = \eta_{\text{Depression},it} + \eta_{\text{Depression},i}$$

$$\text{Anxiety}_{it} = \eta_{\text{Anxiety},it} + \eta_{\text{Anxity},i}$$

$$\text{Subjective Wellbeing}_{it} = \eta_{\text{Subjective Wellbeing},it} + \eta_{\text{Subjective Wellbeing},i}$$

More specifically, the level-1 model defined the within-person mediation model. Between the independent variable and the mediator (the fixed "a" path), the equation was:

$$\eta_{\text{Emotion Dysregulation},it} = a_w \eta_{\text{Dispositional Mindfulness},it} + g_M time_{it} + e_{Mit}.$$

From the mediator to the dependent variables (the fixed "b" paths), the equations were:

$$\eta_{Depression,it} = b_{w1}\eta_{Emotion\ Dysregulation,it} + c'_{w1i}\eta_{Dispositional\ Mindfulness,it} + g_{DV1}time_{it} + e_{DV1it}$$

$$\eta_{Anxiety,it} = b_{w2}\eta_{Emotion\ Dysregulation,it} + c'_{w2}\eta_{Dispositional\ Mindfulness,it} + g_{DV2}time_{it} + e_{DV2it}$$

$$\eta_{Subjective\ Wellbeing,it} = b_{w3}\eta_{Emotion\ Dysregulation,it} + c'_{w3}\eta_{Dispositional\ Mindfulness,it} + g_{DV3}time_{it} + e_{DV3it}$$

Here, $\eta_{Variable,it}$ was the latent within-person component of variable for person $i$ at time $t$. The coefficient $\alpha_w$ was the within-person association between the independent variable and the mediator. The coefficients $b_{w1}$, $b_{w2}$, and $b_{w3}$ quantified the within-person relationships between the mediator and the three mental health outcome variables, i.e., depression symptoms, anxiety symptoms, and subjective well-being, respectively. The coefficients $c'_{w1i}$, $c'_{w2}$, and $c'_{w3}$ represented the relationships between the independent variable and the three dependent variables, respectively, after controlling for the effect of the mediator. The random slope $c'_{w1i}$ was assumed to follow a normal distribution with a mean of $c'_{w10}$ and a variance of $\sigma^2_{w,c}$.

The level-2 model was the between-person mediation model. Specifically, from the independent variable to the mediator (the "a" path at the between-person level), the equation was:

$$\eta_{Emotion\ Dysregulation,i} = \mu_M + a_b\eta_{Dispositional\ Mindfulness,i} + e_{Mi}.$$

From the mediator to the dependent variables (the "b" paths at the between-person level), the equations were:

$$\eta_{Depression,i} = \mu_{DV1} + b_{b1}\eta_{Emotion\ Dysregulation,i} + c'_{b1}\eta_{Dispositional\ Mindfulness,i} + \gamma_{w,1}Gender_{it} + e_{DV1i}$$

$$\eta_{Anxiety,i} = \mu_{DV2} + b_{b2}\eta_{Emotion\ Dysregulation,i} + c'_{b2}\eta_{Dispositional\ Mindfulness,i} + \gamma_{w,2}Gender_{it} + e_{DV2i}$$

$$\eta_{Subjective\ Wellbeing,i} = \mu_{DV3} + b_{b3}\eta_{Emotion\ Dysregulation,i} + c'_{b3}\eta_{Dispositional\ Mindfulness,i} + e_{DV3i}.$$

Similar to the level-1 model, $\eta_{Variable,i}$ was the latent between-person component of a variable for person $i$. The coefficient $\alpha_b$ was the between-person association between the independent variable and the mediator. The coefficients $b_{b1}$, $b_{b2}$, and $b_{b3}$ quantified the between-person relationships between the mediator and the three mental health outcome variables, respectively. The coefficients $c'_{b1}$, $c'_{b2}$, and $c'_{b3}$ represented the between-person relationships between the independent variable and the three dependent variables, respectively, after controlling for the effect of the mediator. Following the traditional steps of mediation analysis [62], we fitted a multilevel mediation model without the mediator to study the overall effect of mindfulness on the outcomes. The model was estimated under missing data theory using all available data [58, ch.9]. Given the limitations listed by Hayes [63], the ratio of the indirect effect to the total effect was not calculated in determining the strength of the indirect effect. That is, this ratio might be out of the expected range between 0 and 1. Also, the estimate of the ratio might be highly unstable from sample to sample. Unless the sample size was fairly large, Hayes [63, p.189] recommended not "having much faith" in this measure.

## Results

Table 1 summarizes the variable means, *SD*s, ranges of the scales, minima, maxima, ICCs, and design effects. Table 2 summarizes the between-person correlations among the variables under study. Specifically, dispositional mindfulness, emotion dysregulation, and mental health

**Table 2. Zero-order correlations of the variables under study (N = 191).**

| Variable | (1) | (2) | (3) | (4) | (5) | (6) | (7) | (8) | (9) | (10) | (11) | (12) | (13) | (14) | (15) | (16) |
|---|---|---|---|---|---|---|---|---|---|---|---|---|---|---|---|---|
| (1) Gender (0 = male; 1 = female) | - | | | | | | | | | | | | | | | |
| **Time 1** | | | | | | | | | | | | | | | | |
| (2) Dispositional mindfulness | .07 | - | | | | | | | | | | | | | | |
| (3) Emotion dysregulation | -.09 | -.55*** | - | | | | | | | | | | | | | |
| (4) Depressive symptoms | -.02 | -.48*** | .69*** | - | | | | | | | | | | | | |
| (5) Anxiety symptoms | -.11 | -.42*** | .65*** | .77*** | - | | | | | | | | | | | |
| (6) Subjective well-being | .11 | .64*** | -.58*** | -.65*** | -.56*** | - | | | | | | | | | | |
| **Time 2** | | | | | | | | | | | | | | | | |
| (7) Dispositional mindfulness | .19* | .58*** | -.54*** | -.41*** | -.37*** | .57*** | - | | | | | | | | | |
| (8) Emotion dysregulation | -.03 | -.42*** | .76*** | .63*** | .59*** | -.49*** | -.62*** | - | | | | | | | | |
| (9) Depressive symptoms | -.05 | -.23** | .49*** | .67*** | .57*** | -.41*** | -.47*** | .63*** | - | | | | | | | |
| (10) Anxiety symptoms | -.09 | -.21** | .39*** | .51*** | .52*** | -.43*** | -.47*** | .52*** | .73*** | - | | | | | | |
| (11) Subjective well-being | .13 | .46*** | -.54*** | -.51*** | -.40*** | .68*** | .70*** | -.62*** | -.52*** | -.52*** | - | | | | | |
| **Time 3** | | | | | | | | | | | | | | | | |
| (12) Dispositional mindfulness | .17* | .53*** | -.47*** | -.42*** | -.38*** | .47*** | .58*** | -.54*** | -.44*** | -.38*** | .51*** | - | | | | |
| (13) Emotion dysregulation | -.12 | -.42*** | .70*** | .55*** | .48*** | -.44*** | -.49*** | .75*** | .52** | .38** | -.53*** | -.65*** | - | | | |
| (14) Depressive symptoms | -.16* | -.45*** | .53*** | .62*** | .57*** | -.44*** | -.44*** | .56*** | .67*** | .54*** | -.47*** | -.61** | .68*** | - | | |
| (15) Anxiety symptoms | -.16* | -.39*** | .44*** | .53*** | .54*** | -.42*** | -.36*** | .48*** | .56*** | .56** | -.44*** | -.49*** | .59** | .80** | - | |
| (16) Subjective well-being | .20* | .50*** | -.43*** | -.45*** | -.38*** | .59*** | .55*** | -.47*** | -.46*** | -.48*** | .69*** | .71*** | -.56*** | -.64*** | -.60*** | - |

*$p < .05$,

**$p < .01$,

***$p < .001$.

outcomes were correlated with each other at $p$s < .05 over time, ranging widely from small to large effect sizes [64]. Gender was associated with T2 and T3 dispositional mindfulness, as well as T3 depressive symptoms, anxiety symptoms, and subjective well-being, $p$s < .05. The variables showed substantial within-person correlations, which were captured by the random intercepts in the multilevel model. Changes in means over time were observed, suggesting the necessity of including time as a covariate.

## Multilevel structural equation modeling

The ICCs of the variables under study ranged from .51 (anxiety symptoms) to .73 (emotion dysregulation), $p$s < .001. This indicated substantial between-person variances, which should be considered through multilevel modeling. The design effect was larger than two for any of the studied variables, except for anxiety symptoms, suggesting that the hierarchical structure of the data should be considered [59]. Based on the R-squared measures for multilevel models proposed by Rights and Sterba [65], we computed the proportion of within-cluster outcome variance explained by level-1 predictors via fixed slopes and random slope variation (i.e., $R_w^{2(f_{1v})}$ in [65]) and the proportion of between-cluster outcome variance explained by level-2 predictors via fixed slopes (i.e., $R_b^{2(f_2)}$ in [65]) for each of the three outcome variables. Specifically, dispositional mindfulness, emotion dysregulation, and time explained 23% of the within-person variances via fixed slopes and random slope variation for depressive symptoms, and 18% and

30% of the within-person variances via fixed slopes for anxiety symptoms and subjective well-being, respectively. At the between-person level, dispositional mindfulness, emotion dysregulation and gender explained 31% and 42% of the between-person variances for depression and anxiety symptoms, respectively. Dispositional mindfulness and emotion dysregulation explained 67% of the between-person variances for subjective well-being (see Fig 1 for details on path coefficients).

**Within-person indirect effect of dispositional mindfulness.** The effect of dispositional mindfulness on depressive symptoms was negatively mediated by emotion regulation, $\widehat{ab} = -.12$, $p < .001$, 95% CI = [-.18, -.06], with a significant mean direct effect, $\widehat{c}'_0 = -.17$, $p < .01$, 95% CI = [-.30, -.05]. Specifically, dispositional mindfulness predicted lower emotion dysregulation, $\widehat{a} = -.31$, $p < .001$, 95% CI = [-.41, -.21]. Emotion dysregulation further predicted greater depressive symptoms, $\widehat{b} = .40$, $p < .001$, 95% CI = [.26, .54]. The results regarding anxiety symptoms showed similar patterns (see Table 3). Specifically, emotion dysregulation inversely predicted by greater dispositional mindfulness was associated with greater anxiety symptoms, $\widehat{ab} = -.15$, $p < .001$, 95% CI = [-.23, -.07], suggesting a partial mediation effect, $\widehat{c}' = -.19$, $p = .01$, 95% CI = [-.34, -.04].

Regarding the indirect effect of dispositional mindfulness on subjective well-being through emotion dysregulation, we found a partial positive mediation effect, $\widehat{ab} = .13$, $p = .002$, 95% CI = [.05, .22]; $\widehat{c}' = .60$, $p < .001$, 95% CI = [.41, .79]. Specifically, dispositional mindfulness was negatively associated with emotion dysregulation, which in turn predicted worse subjective well-being, $\widehat{b} = -.43$, $p < .001$, 95% CI = [-0.20, -.66] (see Table 4).

**Between-person indirect effect of dispositional mindfulness.** The effects of dispositional mindfulness on symptoms of depression and anxiety were found to be negatively mediated by emotion dysregulation (see Table 3). Notably, full mediation effects were found at the between-person level, as indicated by the nonsignificant direct effects of dispositional mindfulness on symptoms of depression, $\widehat{c}' = .46$, $p = .41$, 95% CI = [-.63, 1.54] and anxiety, $\widehat{c}' = -.23$, $p = .15$, 95% CI = [-.54, .08]. Greater dispositional mindfulness was associated with lower emotion dysregulation. Emotion dysregulation, in turn, was associated with greater symptoms of depression and anxiety.

The between-person indirect effect of dispositional mindfulness on subjective well-being was not significant, $\widehat{ab} = .22$, $p = .07$, 95% CI = [-.02, .45]. Although the "a" path between mindfulness and emotion dysregulation was significant, the "b" path between emotion dysregulation and well-being was not, $\widehat{b} = -.24$, $p = .08$, 95% CI = [-.50, .03] (see Table 4).

## Discussion

Building on existing theories on mindfulness, emotion regulation, and well-being [12, 13], this study supported emotion dysregulation as a mediator between dispositional mindfulness and mental health outcomes in university-enrolled Chinese emerging adults. Findings based on multi-level modeling further suggested within- and between-person nuances for the associations (see Fig 1). At the within-person level, emotion dysregulation mediated between dispositional mindfulness and all mental health outcomes. Surprisingly, time was associated with worse emotion dysregulation and well-being, suggesting that psychological functioning worsened as a function of time, potentially due to an increasing level of academic stress over the school year. At the between-person level, dispositional mindfulness negatively predicted emotion dysregulation and positively predicted subjective well-being, whereas emotion dysregulation positively predicted symptoms of depression and anxiety, but not subjective well-being.

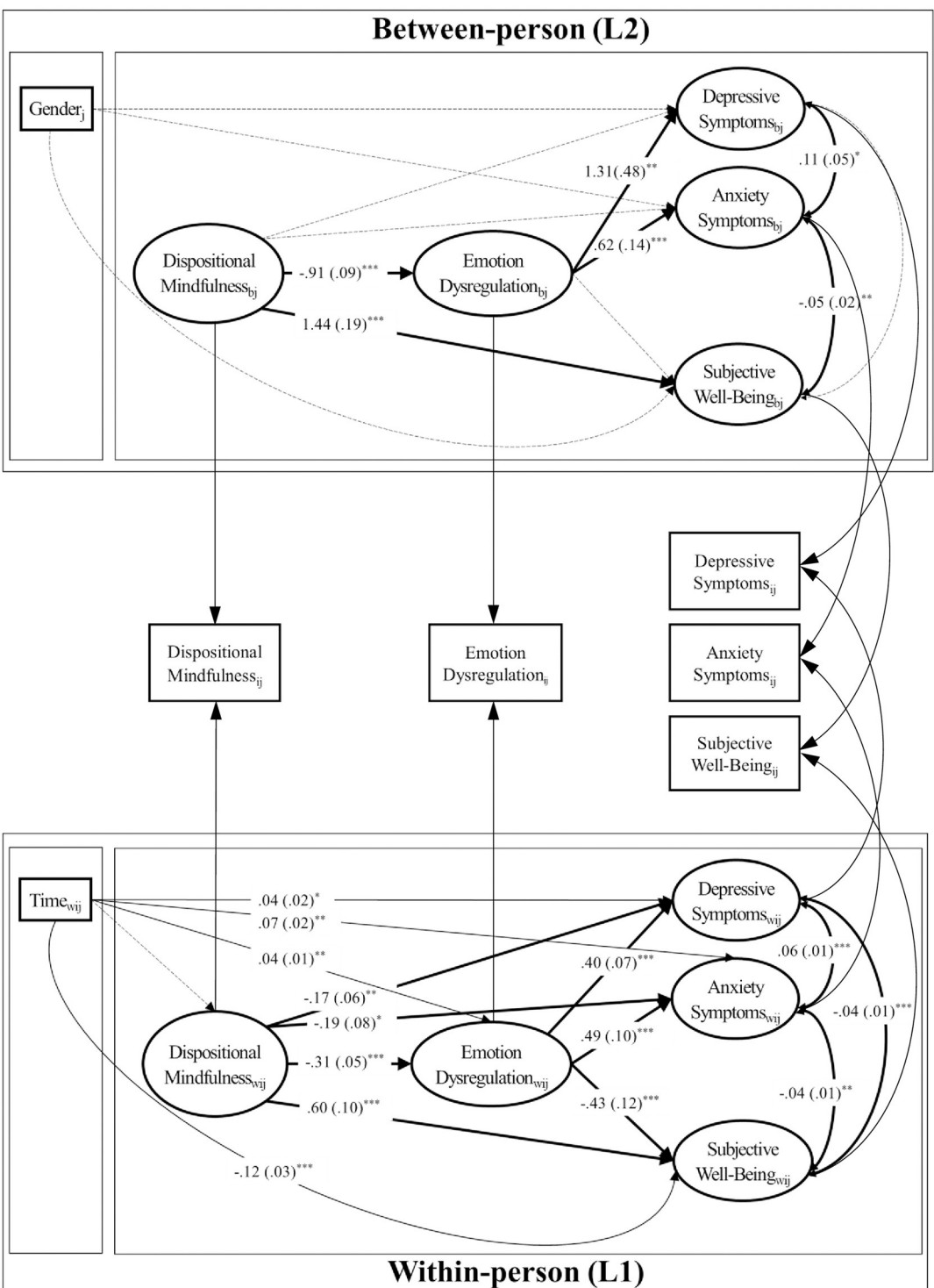

**Fig 1. Multilevel mediation model of dispositional mindfulness, emotion dysregulation, and mental health outcomes.** $^{*}p$ = /< .05, $^{**}p$ = /< .01, $^{***}p$ = /< .001. Unstandardized parameter estimates and standard errors in parentheses are presented. For simplicity, the random slope of depressive symptoms regressed on dispositional mindfulness is not depicted.

**Table 3. Unstandardized parameter estimates and standard errors of the multilevel model.**

| Parameter | Unstandardized B (SE) |
|---|---|
| *Within-Person Effect* | |
| Time | |
| $\rightarrow$ Dispositional mindfulness | -.03 (.02) |
| $\rightarrow$ Emotion dysregulation | .04 (.01)** |
| $\rightarrow$ Depressive symptoms | .04 (.02)* |
| $\rightarrow$ Anxiety symptoms | .07 (.02)** |
| $\rightarrow$ Subjective Well-being | -.12 (.03)*** |
| Dispositional mindfulness | |
| $\rightarrow$ Emotion dysregulation | -.31 (.05)*** |
| $\rightarrow$ Depressive symptoms (mean, $c'_{w10}$) | -.17 (.06)** |
| $\rightarrow$ Anxiety symptoms | -.19 (.08)* |
| $\rightarrow$ Subjective Well-being | .60 (.10)*** |
| Emotion dysregulation | |
| $\rightarrow$ Depressive symptoms | .40 (.07)*** |
| $\rightarrow$ Anxiety symptoms | .49 (.10)*** |
| $\rightarrow$ Subjective well-being | -.43 (.12)*** |
| Depressive symptoms $\longleftrightarrow$ Anxiety symptoms | .06 (.01)*** |
| Depressive symptoms $\longleftrightarrow$ Subjective Well-being | -.04 (.01)*** |
| Anxiety symptoms $\longleftrightarrow$ Subjective Well-being | -.04 (.01)** |
| *Between-Person Effect* | |
| Gender (0 = male; 1 = female) | |
| $\rightarrow$ Depressive symptoms | .03 (.09) |
| $\rightarrow$ Anxiety symptoms | -.07 (.11) |
| Dispositional mindfulness | |
| $\rightarrow$ Emotion dysregulation | -.91 (.09)*** |
| $\rightarrow$ Depressive symptoms | .46 (.55) |
| $\rightarrow$ Anxiety symptoms | -.23 (.16) |
| $\rightarrow$ Subjective Well-being | 1.44 (.19)*** |
| Emotion dysregulation | |
| $\rightarrow$ Depressive symptoms | 1.31 (.48)** |
| $\rightarrow$ Anxiety symptoms | .62 (.14)*** |
| $\rightarrow$ Subjective Well-being | -.24 (.14) |
| Depressive symptoms $\longleftrightarrow$ Anxiety symptoms | .11 (.05)* |
| Depressive symptoms $\longleftrightarrow$ Subjective well-being | .01 (.05) |
| Anxiety symptoms $\longleftrightarrow$ Subjective well-being | -.05 (.02)** |
| Depressive symptoms $\longleftrightarrow$ $c'_{w1i}$ | -.14 (.06)* |
| Anxiety symptoms $\longleftrightarrow$ $c'_{w1i}$ | -.02 (.02) |
| Subjective well-being $\longleftrightarrow$ $c'_{w1i}$ | -.01 (.02) |
| Dispositional mindfulness $\longleftrightarrow$ $c'_{w1i}$ | .01 (.01) |
| Emotion dysregulation $\longleftrightarrow$ $c'_{w1i}$ | -.02 (.02) |
| $c'_{w1i} \longleftrightarrow c'_{w1i}$ | .05 (.02)* |

*$p < .05$,

**$p < .01$,

***$p < .001$. The random coefficient $c'_{w1i}$ represents the within-person relationship between dispositional mindfulness and depressive symptoms, after controlling for the effect of emotion dysregulation. $c'_{w10}$ is the mean of $c'_{w1i}$ across participants.

**Table 4. Results of multilevel mediation analyses concerning the mediating effect of emotion dysregulation between dispositional mindfulness and mental health outcomes.**

| Outcome | a | b | c' | ab | c |
|---|---|---|---|---|---|
| | | | Within-Person | | |
| Depression | | .40[.26,.54] * | -.17[-.30,-.05] * | -.12[-.18,-.06] * | -.29[-.43,-.16] * |
| Anxiety | -.31[-.41,-.21]* | .49[.29,.69] * | -.19[-.34,-.04] * | -.15[-.23,-.07] * | -.34[-.49,-.19] * |
| Subjective well-being | | -.43[-.66,-.20] * | .60[.41,.79] * | .13[.05,.22] * | .73[.52,.94] * |
| | | | Between-Person | | |
| Depression | | 1.31[.36,2.26] * | .46[-.63,1.54] | -1.19[-2.01,-.38] * | -.74[-1.35,-.13] * |
| Anxiety | -.91[-1.09,.74] * | .62[.35,.89] * | -.23[-.54,.08] | -.57[-.85,-.29] * | -.80[-1.01,-.59] * |
| Subjective well-being | | -.24[-.50,.03] | 1.44[1.07,1.82] * | .22[-.02,.45] | 1.66[1.40,1.93] * |

The numbers with asterisk (*) indicate significant results. Numbers within brackets are the lower and upper limits of confidence intervals. The coefficients *a*, *b*, *c*, and *c'* represent the paths from dispositional mindfulness to emotion dysregulation, from emotion dysregulation to mental health outcomes, and from dispositional mindfulness to mental health outcomes before and after controlling for the effect of emotion dysregulation. The coefficient *ab* represents the indirect effect of emotion dysregulation. Note that the within-person *c'* and *c* paths for depression was the average of *c'* and *c* paths across participants. The upper panel reports the results regarding the within-person level whereas the lower displays those for the between-person level.

These findings advanced the field by establishing within- and between-person relations between dispositional mindfulness and mental health outcomes via emotion dysregulation.

The present study used multilevel modeling by partitioning between-person relations from within-person relations. Importantly, the variables were related in the hypothesized directions within-person, suggesting emotion dysregulation as a central intraindividual process linking dispositional mindfulness and mental health outcomes. By orienting to the present moment and paying attention on purpose and non-judgmentally [66], individuals became more mindful and were more capable of decentering themselves from subjective emotional experience [67], less preoccupied with invalidating, avoiding, or rejecting emotional experiences [68], more likely to disengage themselves from autopilot [69], and more likely to engage in adaptive emotion regulation [12, 13]. Based on these findings, practitioners could tailor their practices and interventions to the progress of each client. For example, to enhance subjective well-being and reduce psychological symptoms, practitioners can enhance within-person improvements in mindfulness and emotion regulation, as mindfulness affects mental health outcomes both directly and through adaptive emotion regulation.

Turning to the between-person findings, dispositional mindfulness was directly associated with emotion dysregulation and subjective well-being, but not with symptoms of depression and anxiety. These findings signified that when person *i* had a higher score in dispositional mindfulness than person *j*, then person *i* was also more likely to have a lower score in emotion dysregulation and a higher score in subjective well-being than person *j*. Despite a lack of a direct relation between dispositional mindfulness and psychological symptoms, as an explanatory variable emotion dysregulation accounted for the between-person relation. The findings resonated with recent findings conducted in the Chinese context, in that emotion dysregulation mediated between dispositional mindfulness and symptoms of depression and anxiety [10]. Surprisingly, emotion dysregulation did not hold up as a between-person mediator between dispositional mindfulness and subjective well-being. In fact, emotion dysregulation was not related to subjective well-being at all, after controlling for the effect of dispositional mindfulness. These findings contradicted recent cross-sectional studies [30, 70] showing the link between Chinese college students' emotion regulation strategies and life satisfaction (i.e., a major component of subjective well-being [71]). Instead of a general dysregulation of emotions, perhaps nuances such as emotion regulation strategies (e.g., cognitive reappraisal) [30],

regulatory flexibility [72], and affective states, frequency, and intensity [73] are more proximal predictors of subjective well-being. Alternatively, perhaps over time, emotion dysregulation is more strongly associated with negative outcomes (e.g., psychological symptoms) than with positive outcomes (e.g., positive well-being and satisfaction with life), after controlling for the effect of dispositional mindfulness. Furthermore, stage-salient variables in emerging adulthood, such as achieving financial independence, having greater responsibility, and having greater commitment in romantic relationships [74], might be more strongly associated with positive well-being. Consequently, future studies should replicate the present findings and investigate other longitudinal predictors of subjective well-being in the Chinese context. In terms of clinical implications, practitioners should be made aware of their clients' levels of mindfulness, emotion dysregulation, and mental health relative to other people. For example, returning to Fig 1, in relation to person *j*, person *i*'s greater level of dispositional mindfulness was associated with a lower level of emotion dysregulation and better well-being than person *j*, but not directly with fewer symptoms of depression and anxiety. Understanding these associations could help practitioners gauge the between-person importance or relevance of mindfulness in various mental health outcomes. Finally, gender was not related to psychological symptoms in the model. However, our small sample of men (i.e., 9.95%) precluded us from drawing meaningful conclusions about gender as a correlate of psychological symptoms.

Time was related to greater emotion dysregulation, greater symptoms of depression and anxiety, and worse subjective well-being. Contrary to previous research suggesting that individuals were more capable of emotion regulation as they mature [75], our findings suggested the opposite. It might be that our short-term longitudinal data were insufficient to capture a positive development in emotion regulation. Hence, a long-term longitudinal approach spanning over several years may be necessary to verify whether this is the case. Alternatively, given the participants were emerging adults enrolled in a university, they may be facing increasing stress as they approached the end of the semester or graduation. Future work is needed to rule out confounding variables, such as academic stress and sample characteristics, associated with time and mental health [76].

Moving onto the mean levels, the current university participants reported a similar average of dispositional mindfulness, emotion dysregulation, and symptoms of depression and anxiety compared to others Chinese samples [45, 47, 51]. Paradoxically, our participants also reported a greater level of emotional, psychological, and social well-being than did another college sample [57]. Again, future studies should replicate the mean levels as well as the strength of associations in other Chinese samples.

## Limitations and future directions

The study has several limitations. First, we utilized self-report measures. Future researchers may adopt a multi-method approach by incorporating physiological and neural measures. Next, given this sample comprised mainly of emerging adults at a university, it is uncertain whether the findings can be generalized to other contexts. Future studies should examine the relation in a more representative sample from the community. In addition, the timeframes of the mental health questionnaires differ. For example, the PHQ-9 assessed depressive symptoms in the past two weeks, whereas the MHC-SF assessed emotional, social, and psychological well-being in the past four weeks. Future research may consider standardizing the timeframes to increase precision of the variables. Furthermore, our participants were mainly female. Future studies with gender-balanced samples are necessary to draw meaningful conclusions for the effects of gender. Finally, researchers should translate the present findings and design targeted interventions to improve mental health outcomes.

## Conclusion

This study calls attention to the relation between dispositional mindfulness and mental health outcomes through emotion dysregulation. Taking into account the effect of time, findings based on multilevel modeling demonstrated differential effects at the within- and the between-person levels, thereby suggesting a need to partition these levels in future research. In addition, our findings converged to underscore the association between dispositional mindfulness and different aspects of mental health in emerging adulthood, including symptoms of depression and anxiety, as well as emotional, social, and psychological well-being. Psychological intervention programs and public health campaigns geared toward enhancing mindfulness and emotion regulation merit future research investigations.

## Supporting information

**S1 Data.**
(DAT)

## Acknowledgments

We would like to thank the reviewers for the helpful comments. We would also like to thank Ming Chen, Wing Yee Cheng, and Man Chong Leung for their assistance.

## Author Contributions

**Conceptualization:** Rebecca Y. M. Cheung.

**Data curation:** Rebecca Y. M. Cheung.

**Formal analysis:** Rebecca Y. M. Cheung, Zijun Ke.

**Funding acquisition:** Rebecca Y. M. Cheung, Zijun Ke.

**Investigation:** Rebecca Y. M. Cheung.

**Methodology:** Zijun Ke.

**Project administration:** Rebecca Y. M. Cheung.

**Resources:** Rebecca Y. M. Cheung.

**Writing – original draft:** Rebecca Y. M. Cheung, Zijun Ke, Melody C. Y. Ng.

**Writing – review & editing:** Rebecca Y. M. Cheung, Zijun Ke, Melody C. Y. Ng.

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
