## [Decision Letter · Decision Letter 0]

30 Apr 2020

PONE-D-20-06030

Mindful Awareness and Mental Health: A Multilevel Model with Emotion Regulation as a Mediator

PLOS ONE

Dear Dr Cheung,

Thank you for submitting your manuscript to PLOS ONE. After careful consideration, we feel that it has merit but does not fully meet PLOS ONE’s publication criteria as it currently stands. Therefore, we invite you to submit a revised version of the manuscript that addresses the points raised during the review process.

We would appreciate receiving your revised manuscript by Jun 14 2020 11:59PM. To enhance the reproducibility of your results, we recommend that if applicable you deposit your laboratory protocols in protocols.io, where a protocol can be assigned its own identifier (DOI) such that it can be cited independently in the future. For instructions see: http://journals.plos.org/plosone/s/submission-guidelines#loc-laboratory-protocols

We look forward to receiving your revised manuscript.

Kind regards,

Therese van Amelsvoort

Academic Editor

PLOS ONE

Journal Requirements:

Additional Editor Comments (if provided):

Reviewers' comments:

Reviewer's Responses to Questions

**Comments to the Author**

1. Is the manuscript technically sound, and do the data support the conclusions?

Reviewer #1: Yes

Reviewer #2: Yes

Reviewer #3: Partly

2. Has the statistical analysis been performed appropriately and rigorously? 

Reviewer #1: I Don't Know

Reviewer #2: Yes

Reviewer #3: I Don't Know

3. Have the authors made all data underlying the findings in their manuscript fully available?

Reviewer #1: Yes

Reviewer #2: Yes

Reviewer #3: Yes

4. Is the manuscript presented in an intelligible fashion and written in standard English?

Reviewer #1: Yes

Reviewer #2: Yes

Reviewer #3: Yes

5. Review Comments to the Author

Reviewer #1: Thank you for the interesting manuscript, studying emotion regulation as a mediator between mindfulness and mental health in the Chinese young adult population. I am happy to read that both the within- and between subject factors are taken into consideration in this study design.

A general remark relates to the umbrella term ‘mental health’ used in the research question, this remains rather broad. The introduction includes a description, but throughout it is not always clear how mental health is defined with respect to design and outcomes. The separate measures (depression, anxiety, and subjective wellbeing) are sometimes interchanged with the broader mental health concept. Does this then refer to a composite measure of these questionnaires?

A very high percentage of the study population is female (90%), this is correctly mentioned as a limitation at the end of the manuscript, but briefly. Gender is taken as a covariate in the analyses and conclusions are drawn on these analyses in the beginning of the discussion. This seems to me a big leap and I would advise to be careful with this statement. Reflecting on this earlier in the discussion would help. Another suggestion is to change the number into percentage for easy reading.

In the discussion, more attention could be payed to the implications of the findings for the wellbeing of the Chinese population. Some elements of the introduction, methods and results do not resonate in this last section.

Introduction:

The introduction provides clear evidence for the link between mindfulness, emotion regulation and mental health. The importance of considering all three factors is furthermore evident. I did have some difficulties keeping track of the order in which the evidence is discussed. The three concepts are sometimes mixed within a paragraph aiming to focus on one of the concepts. A clearer distinction would aid readers in the process. The rational for examining both within- and between subject findings simultaneously (an important part of this manuscript) could use some elaboration.

In the first paragraph, I would avoid using the words adjustment outcomes and change these to mental health outcomes to make the link explicit.

Section 'processes of mental health in the Chinese context':

- This section contains several long sentences with unclear structures.

- ‘similar findings were suggested’ needs to be ‘were found’.

- I would highlight a study into emotion regulation in this context to balance the section (e.g., take out a mindfulness example and replace it with one for emotion regulation).

- ‘These findings highlight’ in present tense.

Section 'within- vs. between-subject effects':

- What about emotion regulation in the first sentence?

- ’Partitioning between within- and between-person’: consider using a synonym (e.g. among).

- A clinical interpretation of the importance to distinguish the within-person results and between-person results is lacking. Why is it relevant to separate them, apart from possible biases? What are the differences in interpretation for between-person results and within-person results? Examples are given for within-person results but these are not compared to between-person differences. How can separating both approaches help advance guidelines for future treatment options?

- The transition between the two paragraphs in this section is not very clear.

'This study':

- Is the analysis technique novel or the used approach?

- Why is this unique? Elaborate in the paragraph above.

- The information ‘may be linked to covariates including time and gender’ is new information. A motivation for these choices could be added to the introduction.

Method:

- Participants were at 18-27 years old: needs to be between. This information seems part of the result section (same as the retention rate).

- What were the inclusion and exclusion criteria? Which number of participants was aimed at? How many emails were send out? Some basic Information is lacking.

- The begin structure is unclear; some information from the procedure part belongs to a part about participants and vice versa.

- What sort of informed consent was given?

- How much time did participants investigate in the study?

Measures section:

- This section could benefit from a description of the reliability and validity of the used scales.

- The timeframes of the mental health questionnaires differ (one assesses the past week, another the past month). Could this have influenced the findings in any way?

- Anxiety symptoms: was used to measure anxiety, not depression.

Data analysis section:

- The general writing style of this section is suggestive, which results in unclearity when reading.

- Why are there no details provided on the multilevel mediation model used here (now upon request)? Was growth curve modelling used? Some information would better fit the introduction.

- Please specify the used levels.

- Separate results from methods.

- The section on the mediation path is difficult to follow.

- The last part of this section is especially unclear, does this mean that the used analysis require not much faith? Why is this not mentioned in the discussion?

Results:

- When reading this section I would first expect general sample information, including cores on all outcome measures. Then correlational information.

- Table 1: This table shows many results at the same time. While it seems correct, it might be easier for readers to separate the correlations from the means and standard deviations of the outcomes (make two tables). Before considering correlations, I am curious about how the population scored on the measures. To aid the interpretation, it would be good to mention some reference points for ranges. For example, to interpret the 2.85 score on emotion regulation, I would like to know what this means. What score reflects serious difficulties with emotion regulation?

- Gender does not make sense as M and SD.

- The within-person correlations are not part of the analyses section, a clear rational for these results does not follow naturally.

Section 'multilevel structural equation modeling':

- Are the design effect results reported somewhere?

- Some information in this section would fit the analyses section better.

- Results are better presented numerically, an interpretation of the variances explained suites the discussion (interesting information). They are part of figure 1, might be good to mention this earlier in the section.

Section 'within-person indirect effect of mindfulness awareness (and between)':

- How do the results mentioned here relate to the heading title of indirect effects (they mention direct effects)?

- Table 2: Please explain the arrows and the c’w1i notations in the notes.

- Table 3: The explanation helps. Some of these elements could be taken into the table, providing a heading for within and between analyses results. Consider keeping the format similar to the other tables (e.g., using * instead of boldfaced numbers for significance).

Discussion:

- The first section clearly summarizes the results. This part could benefit from an integrated interpretation of the findings. For example, what could be the reason for the surprising finding of time?

- The statements ‘findings advance the field by using a process-oriented approach’ and ‘demonstrates the utility of multilevel modeling’ could use more explanation.

- The within-person and between-person definitions (when mindfulness of person i increases etc.) can be left out; this is sufficiently clear from the introduction.

- The section on between-person findings with emotion regulation as an explanatory variable is confusing because of the sentence structures.

- Please elaborate on ‘stage-salient variables’.

Reviewer #2: This paper investigates whether emotion regulation is a mediator between mindful awareness and depression, anxiety, and subjective wellbeing in Chinese young adults. Findings suggest that emotion regulation mediates the association between mindful awareness and wellbeing both at the within-person and the between-person level, with the exception that emotion regulation did not mediate the between-person association between mindfulness and subjective wellbeing.

This is an interesting and well-written paper. I only have a few minor comments.

Minor comments

1. I would prefer the term ‘sum score’ or ‘total score’ rather than ‘composite score’ (p. 14) to reflect the total score on a single questionnaire, given that composite score is usually reserved for a score that reflects a combination of scales.

2. “The item scores were summed to form a composite score of depressive symptoms”. However, Table 1 appears to report mean scores rather than sum scores (on a scale from 0 to 3 instead of the original scale from 1 to 4?).

3. For subjective wellbeing, unlike the other scales, it is not explicitly stated how the score was calculated (mean score of the 14 items?).

4. In Table 1 the mean for emotion regulation is 2.97 but in the supplementary data set it appears to be 3.97 if I am not mistaken.

5. Please explain ‘stage-salient variables’ (p. 24). Do have some suggestions as to which stage-salient variables may be relevant to include in future research?

6. The manuscript contains a few typos:

- p. 10 stwell-being

- p. 14 “how often did you feel that that you had ….”

- p. 14 “A sample items included” “

- p. 14 The 7-item Generalized Anxiety Disorder‐7 … was used to measure depressive symptoms”. Depressive symptoms should be anxiety symptoms.

- p. 13 “The 15-item Mindful Attention Awareness Scale …. on a 6-point scale from 1 (almost always) to 5 (almost never)”. This should say ‘a scale from 1 to 6’?

Reviewer #3: The study does not encompass the Emotion Regulation topic but does emotional dysregulation. The implemented measures suggest another title (see attached file). MAAS measures dispositional mindful awareness and DERS assesses emotional dysregulation.

The sample is not enough, gender was not balanced, there was no intervention, the undergraduate sample is too biased for the conclusions (should be tentative rather than so affirmative).

IMuch data were collected and the information about them were scarcy. You collected much data and there was no information about subscales, cutoff (GAD and CES-D). I need more information and explanation of why the addressed these statistical analyses made by the authors. Table 1 is not an ICC (intraclass correlations), it is intercorrelations.

Discussion can be more tentative than affirmative due to data did not allow those conclusions.

see attached file to see more comments in the document

6. PLOS authors have the option to publish the peer review history of their article (what does this mean?). If published, this will include your full peer review and any attached files.

Reviewer #1: No

Reviewer #2: No

Reviewer #3: Yes: José M Mestre

---

## [Author Response · Author response to Decision Letter 0]

17 Jul 2020

We would like to thank the three reviewers for the thoughtful and constructive comments. Below, we outline the revisions that we have made to address these comments. The revisions have also been highlighted in yellow in the revised manuscript for your convenience.

Reviewer #1

General Comments

Comment: 

Thank you for the interesting manuscript, studying emotion regulation as a mediator between mindfulness and mental health in the Chinese young adult population. I am happy to read that both the within- and between subject factors are taken into consideration in this study design.

Our response:

 We thank Reviewer #1 for the positive comments.

Comment:

A general remark relates to the umbrella term ‘mental health’ used in the research question, this remains rather broad. The introduction includes a description, but throughout it is not always clear how mental health is defined with respect to design and outcomes. The separate measures (depression, anxiety, and subjective wellbeing) are sometimes interchanged with the broader mental health concept. Does this then refer to a composite measure of these questionnaires?

Our response:

 Thank you for the comment. Our definition is consistent with the definition of health provided by the World Health Organization [3]. Specifically, we refer mental health to better mental well-being and low levels of psychological symptoms.

Comment:

A very high percentage of the study population is female (90%), this is correctly mentioned as a limitation at the end of the manuscript, but briefly. Gender is taken as a covariate in the analyses and conclusions are drawn on these analyses in the beginning of the discussion. This seems to me a big leap and I would advise to be careful with this statement. Reflecting on this earlier in the discussion would help. Another suggestion is to change the number into percentage for easy reading.

Our response:

 Thank you for the comment. We have now used percentage and number for easy reading (p. 7). In the Discussion, we have removed, “[g]ender was not related to symptoms of depression and anxiety.” We have instead stated (p. 23), “gender was not related to psychological symptoms in the model. However, our small sample of men (i.e., 9.95%) precluded us from drawing meaningful conclusions about gender as a correlate of psychological distress.”

Comment:

In the discussion, more attention could be payed to the implications of the findings for the wellbeing of the Chinese population. Some elements of the introduction, methods and results do not resonate in this last section.

Our response:

 Thank you for the comment. We have now incorporated existing findings of Chinese samples in the Discussion to make the manuscript more coherent. For example, we have added (p. 22), “[t]he findings resonated with recent findings conducted in the Chinese context, in that emotion dysregulation mediated between dispositional mindfulness and symptoms of depression and anxiety [10].” We have also noted (p. 22), “[t]hese findings contradicted recent cross-sectional studies [29,69] showing the link between Chinese college students’ emotion regulation strategies and life satisfaction (i.e., a major component of subjective well-being [70]).” By situating our findings to the Chinese context, we hope our readers are better informed.

Introduction

Comment:

The introduction provides clear evidence for the link between mindfulness, emotion regulation and mental health. The importance of considering all three factors is furthermore evident. I did have some difficulties keeping track of the order in which the evidence is discussed. The three concepts are sometimes mixed within a paragraph aiming to focus on one of the concepts. A clearer distinction would aid readers in the process. The rational for examining both within- and between subject findings simultaneously (an important part of this manuscript) could use some elaboration.

Our response:

 In the revised introduction, we broadly discussed the role of mindfulness in mental health (p. 3) and then moved onto emotion regulation as a predictor of mental health (p. 3). In the next section, we briefly discussed Teper et al.’s theoretical framework [13] and Garland et al.’s framework [21] concerning the mediating role of emotion regulation (p. 4), which was followed by empirical findings (p. 4-5). To situate the study in the Chinese context, we further discussed the findings in Chinese samples (p. 5), with mindfulness as a predictor of emotion regulation and mental health. To ensure that our audience can fully grasp the meanings of within- vs. between-person effects, we discussed the differences, with examples in the context of mindfulness, emotion regulation, and mental health correlates. Of note, partitioning among within- and between-person associations is important, because otherwise the results could sometimes be biased [34] or uninterpretable [35] (p. 6).

We hope that the flow has now provided a clearer picture in supporting the present study. 

Comment:

In the first paragraph, I would avoid using the words adjustment outcomes and change these to mental health outcomes to make the link explicit.

Our response:

We thank the reviewer for the comment. We have revised the introduction by changing “adjustment outcomes” to “mental health outcomes.” 

Introduction: Section 'processes of mental health in the Chinese context'

Comment:

Section 'processes of mental health in the Chinese context': This section contains several long sentences with unclear structures.

Our response:

 The long sentences have now been shortened to increase the readability (p. 5).

Comment:

‘similar findings were suggested’ needs to be ‘were found’.

Our response:

 We have now updated the sentence on page 5. 

Comment:

I would highlight a study into emotion regulation in this context to balance the section (e.g., take out a mindfulness example and replace it with one for emotion regulation).

Our response:

 Thank you for the comment. We have now highlighted a study (p. 5) showing that changes in emotion regulation strategies were associated with changes in changes in depressive symptoms, life satisfaction, and general health over time [29].

Comment:

‘These findings highlight’ in present tense.

Our response:

We have revised the introduction by changing “These findings highlighted” to “These findings highlight” on page 5.

Introduction: Section 'within- vs. between-person effects'

Comment:

What about emotion regulation in the first sentence?

Our response:

Thank you for the comment. The first sentence has now been updated to, “[a] majority of research to-date focuses on between-person associations of mindfulness, emotion dysregulation, and mental health.”

Comment:

’Partitioning between within- and between-person’: consider using a synonym (e.g. among).

Our response:

We have changed “Partitioning between within- and between-person associations” to “Partitioning among within- and between-person associations” on page 6. 

Comment:

A clinical interpretation of the importance to distinguish the within-person results and between-person results is lacking. Why is it relevant to separate them, apart from possible biases? What are the differences in interpretation for between-person results and within-person results? Examples are given for within-person results but these are not compared to between-person differences. How can separating both approaches help advance guidelines for future treatment options?

Our response:

 Thank you for the comment. In the revised introduction, we have highlighted the statistical significance for distinguishing among the within-person and between-person findings (p. 5-6), in that “partitioning between among within- and between-person associations is important, because otherwise the results could be biased [34] or uninterpretable [35].”

Given that the literature has primarily shown between-person effects (p. 5-6), conclusions could only be drawn, specifically, in that when person i has a higher score in mindfulness than person j, then person i would have a higher score in emotion regulation and mental health than would person j. In this study, we argued that it is theoretically important to also address whether increases in mindfulness in person i are linked to his or her own better emotion regulation and mental health over time. 

The present findings have important implications, as addressed in the Discussion. At the within-person level (p. 21-22), practitioners could tailor their practices and interventions to the progress of each client. For example, to enhance subjective well-being and reduce psychological symptoms, practitioners can enhance within-person improvements in mindfulness and emotion regulation, as mindfulness affects mental health outcomes both directly and through adaptive emotion regulation. At the between-person level (p. 23), practitioners should be made aware of their clients’ levels of mindfulness, emotion dysregulation, and mental health relative to other people. For example, returning to Figure 1, in relation to person j, person i’s greater level of mindfulness was associated with a lower level of emotion dysregulation and better well-being than person j, but not directly with fewer symptoms of depression and anxiety. Understanding these associations could help practitioners gauge the between-person importance or relevance of mindfulness in various mental health outcomes.”

Comment:

The transition between the two paragraphs in this section is not very clear.

Our response:

 Under the section 'Within- vs. Between-Person Effects, we have now strengthened the transition by adding the sentence, “[s]everal studies to-date have demonstrated the utility of within-person analyses in mindfulness and well-being.”

Introduction: 'This study'

Comment:

Is the analysis technique novel or the used approach?

Our response:

 We have deleted “novel” from “This Study” given that it has been used in previous research.

Comment:

Why is this unique? Elaborate in the paragraph above.

Our response:

 In the revised manuscript, we have added (p. 6-7), “[a]s stated earlier, previous research predominantly examined between-person effects among mindfulness, emotion regulation, and mental health [5-11]. Although some studies have begun to examine within-person effects [36-38], few, if any, have teased apart within-person from between-person effects.”

Comment:

The information ‘may be linked to covariates including time and gender’ is new information. A motivation for these choices could be added to the introduction.

Our response:

 Thank you for the comment. On page 5, we have strengthened our arguments concerned with time by adding, “changes in mindful awareness were associated with subsequent changes of stress response and anxiety symptoms [28]. Similarly, changes in emotion regulation strategies were associated with changes in changes in depressive symptoms, life satisfaction, and general health over time [29]. As such, timing and changes were crucial in linking mindfulness, emotion regulation, and mental health outcomes.” 

 As for gender, previous research conducted with Chinese samples revealed that gender was associated with depression [39], anxiety [40], and subjective well-being [41]. As such, gender was included in this study as a covariate (p. 7).

Method

Comment:

Participants were at 18-27 years old: needs to be between. This information seems part of the result section (same as the retention rate).

Our response:

 We have updated “[p]articipants were at 18-27 years old” to “[p]articipants were between 18 and 27 years old.” Since age and retention were not the study focus or part of the hypothesis, we have only included the information under “Participants” of the Method section (p. 7-8).

Comment:

What were the inclusion and exclusion criteria? 

Our response:

 We have added the following sentence (p.8), “Inclusion criteria included college-enrolled emerging adults who were proficient in Chinese, within the age range between 18 and 29 years old [42], who agreed to participate for three time points over the course of a year.”

Comment:

Which number of participants was aimed at? 

Our response:

Thank you for the comment. We have conducted a post hoc power analysis based on Monte Carlo simulations [76]. Based on the estimated parameter values from the multilevel mediation analysis, we repeatedly simulated 1000 new data sets and fitted the same multilevel mediation model to these simulated data sets. By counting the proportion of significant results, we obtained the empirical post hoc power. The results showed that on average, the post hoc power for the six indirect effects considered in the study (three at the within-person level and three at the between-person level) was 80.2%, suggesting that it was adequate to aim for 191 participants.

Reference:

76. Gelman A, Hill J. Data analysis using regression and multilevel/hierarchical models. Cambridge: Cambridge University Press; 2006.

Comment:

How many emails were send out?

Our response:

 Two mass emails were sent out (p. 7-8).

Comment:

The begin structure is unclear; some information from the procedure part belongs to a part about participants and vice versa.

Our response:

 We have worked to ensure that the Participants and Procedures sections were independent from each other (p. 7-8). Specifically, the Participants section involves recruitment method, inclusion criteria, retention rate, and demographics. The Procedures section involves ethics approval information, informed consent details, and data collection procedures.

Comment:

What sort of informed consent was given?

Our response:

We have revised the Procedures section by stating the following (p. 8), “The study was approved by the Human Research Ethics Committee of the first author’s university prior to its implementation. All procedures performed were in accordance with the ethical standards of the institutional research committee and the 1964 Helsinki declaration and its later amendments. Informed consent to participate in the present study was obtained prior to the administration of the questionnaire.” 

Comment:

How much time did participants investigate in the study?

Our response:

 We have now included the following information (p. 8), “[e]ach packet took approximately 30 minutes to complete.”

Method: Measures

Comment:

This section could benefit from a description of the reliability and validity of the used scales.

Our response:

 Based on previous research, we have now included the reliability and validity of the scales used in the present study (p. 8-12).

Comment:

The timeframes of the mental health questionnaires differ (one assesses the past week, another the past month). Could this have influenced the findings in any way?

Our response:

 We agree with Reviewer #1 that the timeframes of the mental health questionnaire differ. As a limitation, we have now stated (p.24), “the timeframes of the mental health questionnaires differ. For example, the PHQ-9 assessed depressive symptoms in the past two weeks, whereas the Mental Health Continuum Short Form assessed emotional, social, and psychological well-being in the past four weeks, whereas the MHC-SF assessed emotional, social, and psychological well-being in the past four weeks. Future research may consider standardizing the timeframes to increase precision of the variables.”

Comment:

Anxiety symptoms: was used to measure anxiety, not depression.

Our response:

Thank you for the comment. We have made relevant changes in the revised manuscript (p. 11).

Method: Data analysis

Comment:

The general writing style of this section is suggestive, which results in unclearity when reading.

Our response:

 We have expanded and revised the Data Analysis section. A more affirmative writing style has been used.

Comment:

Why are there no details provided on the multilevel mediation model used here (now upon request)? Was growth curve modelling used? Some information would better fit the introduction.

Our response:

Thank you for the comment. We have now provided a statistical description of the multilevel mediation model (p. 12-15). The multilevel mediation model used in the analysis was not a growth curve model. To avoid confusion, we have removed sentences related to growth curve modeling in the revised manuscript. Two major differences between multilevel mediation model and growth curve model are as follows:

First, in our analysis, the relationship between the independent variables and the mediator (i.e., reflected by the “a” path) and the relationship between the mediator and the dependent variables (i.e., reflected by the “b” paths) were estimated simultaneously. In growth curve modeling with time-varying covariates, however, these relationships were estimated separately. The estimated “a” path and “b” paths could be correlated when data are nonnormal and when robust methods are used (i.e., when MLR is used in Mplus; see the discussion in http://www.statmodel.com/discussion/messages/11/9365.html?1396831771). In our analysis, MLR was used and the possible association between the estimated “a” path and “b” path could be captured. In contrast, the association could not be captured using growth curve modeling. This might have an impact on the inferences on the indirect effect, which was defined as the product of the “a” path and the “b” path (e.g., see [77; p. 92]; and the explanation regarding the difference between the delta method used in Mplus and Sobel test in http://www.statmodel.com/discussion/messages/11/9365.html?1396831771).

Second, we relied on the multilevel SEM approach [34] to separate the within- and between-person components of variables. In contrast, as a typical multilevel modeling, growth curve modeling relies on group centering to fulfill this objective. As discussed in [34], the multilevel approach was relatively more biased. 

Reference:

77. MacKinnon DP. Introduction to statistical mediation analysis (Multivariate applications series). Taylor & Francis Group/Lawrence Erlbaum Associates; 2008.

Comment:

Please specify the used levels.

Our response:

 We have now specified the used levels in greater detail. 

Comment:

Separate results from methods.

Our response:

Thank you for the comment. The results have been separated from methods.

Comment:

The section on the mediation path is difficult to follow.

Our response:

Thank you for the comment. We have revised the Data Analysis section. Now the logic of this section flows more naturally and it is easier to understand. Specifically, we first described the statistical approach to multilevel mediation used in this study (p. 12). We then explained what covariates and why they were considered in the model (p. 12). Next, we explained why some paths were fixed and some was allowed to be random across participants (p.13). We also described in detail the multilevel mediation model used in the analysis and explained the practical meanings of relevant model parameters (p. 13-15). Finally, we explained why the proportion of indirect effect was not calculated (p. 15).

Comment:

The last part of this section is especially unclear, does this mean that the used analysis require not much faith? Why is this not mentioned in the discussion?

Our response:

 Thank you for the comment. The last sentence of the Data Analysis section (p. 15), “[u]nless the sample size was fairly large, Hayes [60, p.189] recommended not ‘having much faith’ in this measure” explains the reason why we did not compute the ratio of indirect effect to the total effect. It was not related to the method used in the analysis.

Results

Comment:

When reading this section I would first expect general sample information, including cores on all outcome measures. Then correlational information... Before considering correlations, I am curious about how the population scored on the measures. To aid the interpretation, it would be good to mention some reference points for ranges. For example, to interpret the 2.85 score on emotion regulation, I would like to know what this means. What score reflects serious difficulties with emotion regulation?

Our response:

Thank you for the comment. We have now briefly described the correlational findings at the beginning of the results section (p. 15).

As for how the current sample scored on the measures compared to other study samples, we have now included the information under the measures section (p. 8-12). Specifically, compared to other Chinese samples, participants from this study had similar mean levels of dispositional mindfulness [44], emotion dysregulation [46], and symptoms of depression and anxiety [50]. Nevertheless, they had greater mean levels of subjective well-being [56]. This information has also been included in the discussion section (p. 24). 

Comment:

Table 1: This table shows many results at the same time. While it seems correct, it might be easier for readers to separate the correlations from the means and standard deviations of the outcomes (make two tables). 

Our response:

We have separated Table 1 into two tables according to Reviewer #1’s recommendation. Table 1 shows the means, standard deviations, ranges, ICCs, and design effects, whereas Table 2 shows the zero-order correlations. 

Comment:

Gender does not make sense as M and SD.

Our response:

We have removed the M and SD of gender in Table 1. 

Comment:

The within-person correlations are not part of the analyses section, a clear rational for these results does not follow naturally.

Our response:

Thank you for the comment. We have revised the Data Analysis section (p. 12-15). Now the subsection contains a detailed description on the multilevel mediation model. The analyses regarding the within- and between-person associations have also been described.

Results: 'Multilevel Structural Equation Modeling'

Comment:

Are the design effect results reported somewhere?

Our response:

The design effect results have now been reported in Table 1.

Comment:

Some information in this section would fit the analyses section better.

Our response:

 To enhance clarity and organization, we have restructured the Data Analysis and the Results sections (p. 12-20).

Comment:

Results are better presented numerically, an interpretation of the variances explained suites the discussion (interesting information). They are part of figure 1, might be good to mention this earlier in the section.

Our response:

 Thank you for the comment. As we were unable to denote the % of variance in Figure 1, we did not describe the figure early on. However, at the end of the paragraph, we have recommended the readers to refer to Figure 1 for specific path coefficients.

Results: 'Within-person indirect Effect of dispositional mindfulness (and between)'

Comment:

How do the results mentioned here relate to the heading title of indirect effects (they mention direct effects)?

Our response:

 Thank you for the comment. The section refers to indirect effects.

Comment:

Table 3: Please explain the arrows and the c’w1i notations in the notes.

Our response:

Thank you for the comment. We have added an explanation in note on Table 3 to explain the notations.

Comment:

Table 4: The explanation helps. Some of these elements could be taken into the table, providing a heading for within and between analyses results. Consider keeping the format similar to the other tables (e.g., using * instead of boldfaced numbers for significance).

Our response:

We have revised Table 4 by using * instead of boldfaced numbers for significant numbers. 

Discussion

Comment:

The first section clearly summarizes the results. This part could benefit from an integrated interpretation of the findings. For example, what could be the reason for the surprising finding of time?

Our response:

 Thank you for the comment. At the beginning of the Discussion, we have included a potential reason for the surprising finding of time (p. 21) by stating, “potentially due to an increasing level of academic stress over the school year.” To ensure that the opening paragraph is clear and succinct, we have explained the findings in greater detail in the subsequent paragraphs.

Comment:

The statements ‘findings advance the field by using a process-oriented approach’ and ‘demonstrates the utility of multilevel modeling’ could use more explanation.

Our response:

 Thank you for the comment. To ensure that our discussion is clear and informative, the statement, “findings advance the field by using a process-oriented approach,” have been revised to “[t]hese findings advanced the field by establishing within- and between-person relations between dispositional mindfulness and mental health via emotion dysregulation.” (p.21)

 The statement, “demonstrates the utility of multilevel modeling” have been revised to “[t]his study used multilevel modeling by partitioning between-person relations from within-person relations.”(p.21)

Comment:

The within-person and between-person definitions (when mindfulness of person i increases etc.) can be left out; this is sufficiently clear from the introduction.

Our response:

We thank Reviewer #1 for the recommendation. We have removed the within-person definition in the discussion. To concretely explain the between-person findings and their clinical significance, however, we have decided to keep the notations (e.g., person i vs person j; p. 22-23). 

Comment:

The section on between-person findings with emotion regulation as an explanatory variable is confusing because of the sentence structures.

Our response:

 We have now revised the section (p.22-23). 

Comment:

Please elaborate on ‘stage-salient variables’.

Our response:

Thank you for the comment. Stage-salient variables refer to variables specific to a certain developmental period, i.e., in our case, emerging adulthood. In the discussion, we have added, “other stage-salient variables in emerging adulthood, such as having increasing responsibility, becoming more financial independent, and having a greater commitment in romantic relationships [73], may be more strongly associated with positive well-being.” (p.23)

Reviewer #2

Comment:

This paper investigates whether emotion regulation is a mediator between mindful awareness and depression, anxiety, and subjective wellbeing in Chinese young adults. Findings suggest that emotion regulation mediates the association between mindful awareness and wellbeing both at the within-person and the between-person level, with the exception that emotion regulation did not mediate the between-person association between mindfulness and subjective wellbeing.

This is an interesting and well-written paper. I only have a few minor comments.

Our response:

We thank Reviewer #2 for the positive comments.

Minor Comments

Comment:

I would prefer the term ‘sum score’ or ‘total score’ rather than ‘composite score’ (p. 14) to reflect the total score on a single questionnaire, given that composite score is usually reserved for a score that reflects a combination of scales.

Our response:

 We have revised the method by changing “composite scores” to “mean scores.” (p.8-12)

Comment:

“The item scores were summed to form a composite score of depressive symptoms”. However, Table 1 appears to report mean scores rather than sum scores (on a scale from 0 to 3 instead of the original scale from 1 to 4?).

Our response:

 Thank you for the comment. We apologize for the confusion. In this study, participants rated on a scale from 0 to 3 for each item of the PHQ-9. Afterwards, the item scores were averaged to form a score. The description has now been updated in the manuscript (p. 10).

Comment:

For subjective wellbeing, unlike the other scales, it is not explicitly stated how the score was calculated (mean score of the 14 items?).

Our response:

 We have now stated how the subjective well-being score was calculated (p. 12), “The item scores were averaged to form a mean score, with higher scores indicating better well-being.” 

Comment:

In Table 1 the mean for emotion regulation is 2.97 but in the supplementary data set it appears to be 3.97 if I am not mistaken.

Our response:

 Thank you for pointing this out. We have corrected the means and standard deviations for emotion dysregulation.

Comment:

Please explain ‘stage-salient variables’ (p. 24). Do have some suggestions as to which stage-salient variables may be relevant to include in future research?

Our response:

Thank you for the comment. Stage-salient variables refer to variables specific to a certain developmental period, i.e., in our case, emerging adulthood. In the discussion, we have added (p.23), “other stage-salient variables in emerging adulthood, such as having increasing responsibility, becoming more financial independent, and having a greater commitment in romantic relationships [73], may be more strongly associated with positive well-being.”

6. The manuscript contains a few typos:

- p. 10 stwell-being

- p. 14 “how often did you feel that that you had ….”

- p. 14 “A sample items included” “

- p. 14 The 7-item Generalized Anxiety Disorder‐7 … was used to measure depressive symptoms”. Depressive symptoms should be anxiety symptoms.

- p. 13 “The 15-item Mindful Attention Awareness Scale …. on a 6-point scale from 1 (almost always) to 5 (almost never)”. This should say ‘a scale from 1 to 6’?

Our response:

 Thank you for the comment. The typos have been removed in the revised manuscript. 

Reviewer #3: 

Title Page

Comment:

Title arose some expectations that were not accomplished later.

Our response:

 Thank you for the comment. We have updated the title to, “Dispositional Mindfulness and Mental Health in Chinese Emerging Adults: A Multilevel Model with Emotion Dysregulation as a Mediator.”

Comment:

DERS: Difficulties in ER Scale measures emotional dysregulation rather than emotion regulation. Dysregulation: A failure to regulate properly.

Our response:

 We agree with Reviewer #3’s comment. In our original manuscript, the scores of the negative items of DERS were reversed, such that the final scores reflected a greater ability in emotion regulation. 

Based on Reviewer #3’s comment, we have unreversed the scores, such that higher scores reflected greater emotion dysregulation. Where appropriate, we have now used “emotion dysregulation” in the revised manuscript.

Comment:

Many authors have pointed out that these mindful scales are not measuring mindful activities, they assess dispositional mindful activities.

Our response:

 We agree with Reviewer #3’s comment and have changed our wordings to “dispositional mindfulness” in describing the variable under study.

Abstract

Comment:

Provide here M and SD for participants’ age.

Our response:

We have revised the abstract by adding M and SD of the participants’ age. 

Comment:

A questionnaire about what?

Our response:

 We have added that the questionnaire assessed their dispositional mindfulness, emotion dysregulation, and mental health outcomes.

Introduction

Comment:

Avoid using regulating emotions in the definition of emotion regulation.

Our response:

 We have revised the revised the sentence to the following, “[e]motion regulation is defined as the process in modulating emotions and emotional responses [14-15].” (p.3)

Introduction: Within- vs. Between-Person Effects

Comment:

Agree but this section above is a little bit tangential to your research. You did not implement a RCT MBI.

Our response:

 We apologize about the confusion and agree with Reviewer #3 that we did not implement a mindfulness-based intervention. Given that very few, if any, studies assessed within- vs. between-person effects of dispositional mindfulness on mental health, we have reviewed mindfulness-based intervention studies using a multilevel modeling approach.

Method

Comment:

Does this Committee a code?

Our response:

We have revised the Method by stating (p. 8), “The study was approved by the Human Research Ethics Committee of the first author’s university institution prior to its implementation. All procedures performed were in accordance with the ethical standards of the institutional research committee and the 1964 Helsinki declaration and its later amendments.”

Comment:

The sample is not enough, gender was not balanced, there was no intervention…

Our response:

 We agree with Reviewer #3 that the sample size could have been larger and gender could be more balanced.

In terms of sample size, we have conducted a post hoc power analysis based on Monte Carlo simulations [76]. Based on the estimated parameter values from the multilevel mediation analysis, we repeatedly simulated 1000 new data sets and fitted the same multilevel mediation model to these simulated data sets. By counting the proportion of significant results, we obtained the empirical post hoc power. The results showed that on average, the post hoc power for the six indirect effects considered in the study (three at the within-person level and three at the between-person level) was 80.2%, suggesting that it was adequate to aim for 191 participants.

As for gender, we have stated in the Discussion (p. 23), “our small sample of men (i.e., 9.95%) precluded us from drawing meaningful conclusions about gender as a correlate of psychological distress.” We have also acknowledged, (p. 24), “our participants were mainly female. Future studies with gender-balanced samples are necessary to draw meaningful conclusions for the effects of gender.”

We understand that intervention studies are necessary to improve mental health. However, it is beyond the scope of the present study. As a future direction, we have added (p. 24), “researchers should translate the present findings and design targeted interventions to improve mental health outcomes.”

Reference:

76. Gelman A, Hill J. Data analysis using regression and multilevel/hierarchical models. Cambridge: Cambridge University Press; 2006.

Comment:

This tool measures dispositional mindful awareness.

Our response:

 We have now updated the description to the following (p.8), “[t]he 12-item Cognitive and Affective Mindfulness Scale – Revised (CAMS-R) [43] was used to assess dispositional mindfulness on a 4-point scale from 1 (rarely/ not at all) to 4 (almost always).” The measure, CAMS-R, broadly assessed dispositional mindfulness, in addition to dispositional mindful awareness.

Comment:

Cronbach's alpha of subscales should be also reported. You can use intercorrelations at T1, T2 and T3 as well.

Our response:

 Thank you for the comment. We have now indicated the Cronbach's alpha of subscales for the Cognitive and Affective Mindfulness Scale – Revised (p. 9), the Difficulties in Emotional Regulation Scale (p. 9), and the Mental Health Continuum Short Form (p. 12). The intercorrelations between T1, T2 and T3 have been reported for all measures as well (p. 8-12).

Comment:

Is there a cut-off in the scoring to indicate depression?

Our response:

 Thank you for the question. The cut-off scores for the Patient Health Questionnaire-9 and the Generalized Anxiety Disorder‐7 measures have now been reported on page 10.

Results

Comment:

Gender: 0 male and 1 female?

Our response:

 We have now denoted “0=male; 1=female” on Tables 1-3.

Comment:

Why are the correlations of mindfulness moderate / low at T1, T2, T3?

Our response:

 The correlations of dispositional mindfulness between T1 and T3 were .53 - .58 over time, ps < .001, suggesting moderate effect sizes [62]. Even though the scale was intended to measure stable and dispositional mindfulness, our data suggest potential changes over time. 

Comment:

Based on the correlations, gender could be a moderator.

Our response:

 We agree with Reviewer #3’s comment. However, our small sample of men (i.e., 9.95%) precluded us from drawing meaningful conclusions about gender as a moderator. As a limitation, we have stated (p. 24), “our participants were mainly female. Future studies with gender-balanced samples are necessary to draw meaningful conclusions for the effects of gender.”

Comment:

M and SD seem to be averaged, this should be notified in the table and put in the first column the range of every single instrument.

Our response:

 We have now stated (p. 15), “Table 1 summarizes the means, SDs, ranges, ICCs, and design effects.”

Comment:

ICC is more appropriate to assess the level of agreement among raters. Where is reported the ICC, Table 1 is not a ICC.

Our response:

Thank you for the comment. We have added ICCs in Table 1 for all the variables under study.

Discussion

Comment:

You did not implement an intervention... The discussion needs to follow a tentative argument… The undergraduate sample is too biased for the conclusions (should be tentative rather than so affirmative). Discussion can be more tentative than affirmative due to data did not allow those conclusions.

Our response:

 Thank you for the comment. We agree with Reviewer #3 that the present study involved no experiments and interventions. Thus, in the discussion, we have avoided wordings such as “causes”, “affects”, or “influences.” Based on the present findings, tentative arguments have been made, without biases or exaggerations. As a limitation, we have added, “researchers should translate the present findings and design targeted interventions to improve mental health outcomes.”

---

## [Decision Letter · Decision Letter 1]

11 Aug 2020

PONE-D-20-06030R1

Dispositional Mindfulness and Mental Health in Chinese Emerging Adults: A Multilevel Model with Emotion Dysregulation as a Mediator

PLOS ONE

Dear Dr. Ke

Thank you for submitting your manuscript to PLOS ONE. After careful consideration, we feel that it has merit but does not fully meet PLOS ONE’s publication criteria as it currently stands. Therefore, we invite you to submit a revised version of the manuscript that addresses the points raised during the review process.

We look forward to receiving your revised manuscript.

Kind regards,

Therese van Amelsvoort

Academic Editor

PLOS ONE

Reviewers' comments:

Reviewer's Responses to Questions

**Comments to the Author**

1. If the authors have adequately addressed your comments raised in a previous round of review and you feel that this manuscript is now acceptable for publication, you may indicate that here to bypass the “Comments to the Author” section, enter your conflict of interest statement in the “Confidential to Editor” section, and submit your "Accept" recommendation.

Reviewer #2: All comments have been addressed

Reviewer #3: All comments have been addressed

2. Is the manuscript technically sound, and do the data support the conclusions?

Reviewer #2: Yes

Reviewer #3: Yes

3. Has the statistical analysis been performed appropriately and rigorously? 

Reviewer #2: Yes

Reviewer #3: Yes

4. Have the authors made all data underlying the findings in their manuscript fully available?

Reviewer #2: No

Reviewer #3: Yes

5. Is the manuscript presented in an intelligible fashion and written in standard English?

Reviewer #2: No

Reviewer #3: Yes

6. Review Comments to the Author

Reviewer #2: The authors have addressed all my comments adequately. However, there are a few issues with the revised version that should be addressed:

1. Table 1. Means and standard deviations of the study variables (N = 191): The table not only includes means and standard deviations but also range, ICC and design effect. Under the column range, however, only a single number is given rather than a range. Furthermore, the abbreviation ICC should be explained in the caption of the table and it should also explain what 'design effect' is.

2. "Based on a cutoff score of 11 for the detection of depressive disorders [49], upon rescaling, our participants’ summed scores of 5.44 - 6.67 between T1 and T3 were below the clinical cutoff": the rescaling is not explained. How were the the PHQ-9 scores rescaled? The same applies for the anxiety symptoms measures with the GAD-7 ("Based on a cutoff score of 10 for the detection of generalized anxiety disorder [51], upon rescaling our participants’ summed scores between 4.22 and 5.67 between T1 and T3 were below the clinical cutoff").

3. "The design effect, defined as 1+ (average cluster size-1)×ICC". This should be explained in the Methods (and in the caption of Table 1 if you include the design effect in this table) rather than introducing it in the Results. In the current manuscript the text first refers to the design effect (first paragraph results, Table 1) and only explains it later.

4. “.... Hayes [60, p.189] recommended ...’”. The reference should be reference 61 rather than 60.

5. The manuscript needs some language editing. E.g.:

- ".... changes in emotion regulation strategies were associated with changes in changes in depressive symptoms ...";

- "Such an increase was found to be more significantly than did participants who ...";

- ".... confounding between-person factors from the within-person associations ..."

Reviewer #3: Congratulations on the implemented efforts to improve the paper. I left some comments throughout the paper to amend some issues that I found (even the reference list)

7. PLOS authors have the option to publish the peer review history of their article (what does this mean?). If published, this will include your full peer review and any attached files.

Reviewer #2: No

Reviewer #3: **Yes: **Jose M Mestre PhD. Department of Psychology, Universidad de Cádiz

---

## [Author Response · Author response to Decision Letter 1]

13 Aug 2020

Reviewer #2

Comment:

Table 1. Means and standard deviations of the study variables (N = 191): (a) The table not only includes means and standard deviations but also range, ICC and design effect. (b) Under the column range, however, only a single number is given rather than a range. (c) Furthermore, the abbreviation ICC should be explained in the caption of the table and it should also explain what 'design effect' is.

Our response:

 Thank you for the comment.

(a) We have updated Table 1’s caption (p. 10) to “Table 1. Descriptive statistics of the variables under study.” 

(b) The range was originally calculated by the difference between lower and higher scores in every scale. To improve clarity, we have relabeled the column to “Range of Scale” and provided the range. To include further descriptive statistics, we have also added “Minimum” and “Maximum” in Table 1.

(c) The abbreviation ICC has now been replaced by “Intraclass Correlation.” A note has been added to the bottom of Table 1 to define “design effect.”

Comment:

"Based on a cutoff score of 11 for the detection of depressive disorders [49], upon rescaling, our participants’ summed scores of 5.44 - 6.67 between T1 and T3 were below the clinical cutoff": the rescaling is not explained. How were the the PHQ-9 scores rescaled? The same applies for the anxiety symptoms measures with the GAD-7 ("Based on a cutoff score of 10 for the detection of generalized anxiety disorder [51], upon rescaling our participants’ summed scores between 4.22 and 5.67 between T1 and T3 were below the clinical cutoff").

Our response:

 Thank you for the comment. Under the PHQ-9 and GAD-7, we have added “upon rescaling the mean scores to summed scores” to clarify how we rescaled the scores.

Comment:

"The design effect, defined as 1+ (average cluster size-1)×ICC". This should be explained in the Methods (and in the caption of Table 1 if you include the design effect in this table) rather than introducing it in the Results. In the current manuscript the text first refers to the design effect (first paragraph results, Table 1) and only explains it later.

Our response:

 Thank you for the comment. We have revised Table 1 (p. 10) and the Methods accordingly (p. 12).

Comment:

“.... Hayes [60, p.189] recommended ...’”. The reference should be reference 61 rather than 60.

Our response:

 The relevant correction has now been made (p. 15 and 34).

Comment:

The manuscript needs some language editing. E.g.:

- ".... changes in emotion regulation strategies were associated with changes in changes in depressive symptoms ...";

- "Such an increase was found to be more significantly than did participants who ...";

- ".... confounding between-person factors from the within-person associations ..."Our response:

 Thank you for the comment. We have conducted language editing throughout the manuscript (e.g., p. 5, 6).

Reviewer #3

Comment: 

Table 1 should include descriptive measures of the study, not just M and SD.

Our response:

 Thank you for the comment. We have now included the means, standard deviations, maxima, minima, ranges of the scales, intraclass correlations, and design effects in Table 1 (p. 10).

Comment: 

In Table 1, “*” should be added to describe ICC significance (e.g., *p < .05, **p < .01, ***p < .001.). Also, add info regarding how you calculated design effect.

Our response:

 Thank you for the comment. We have added “*” to indicate the significance of the ICC (p. 10 and 18). We have also added a note at the bottom of Table 1 to define design effect (p. 10).

Comment: 

Under “Data Analysis,” replace the word “mindfulness” by “dispositional mindfulness.”

Our response:

 Thank you for the comment. The correction has been made (p. 12).

Comment: 

For reference #54, include the retrieving day. Sometimes, the link is no longer available.

Our response:

 Thank you for the comment. We have updated the link and added a retrieving day (p. 33).

---

## [Editor Report · Decision Letter 2]

10 Sep 2020

Dispositional Mindfulness and Mental Health in Chinese Emerging Adults: A Multilevel Model with Emotion Dysregulation as a Mediator

PONE-D-20-06030R2

Dear Dr. Ke,

We’re pleased to inform you that your manuscript has been judged scientifically suitable for publication and will be formally accepted for publication once it meets all outstanding technical requirements.

Kind regards,

Therese van Amelsvoort

Academic Editor

PLOS ONE
---

## [Editor Report · Acceptance letter]

10 Nov 2020

PONE-D-20-06030R2 

Dispositional Mindfulness and Mental Health in Chinese Emerging Adults: A Multilevel Model with Emotion Dysregulation as a Mediator 

Dear Dr. Ke:

I'm pleased to inform you that your manuscript has been deemed suitable for publication in PLOS ONE. Congratulations! Your manuscript is now with our production department. 

Kind regards, 

on behalf of

Prof. Therese van Amelsvoort 

Academic Editor

PLOS ONE